# Metabolite Profile and Antioxidant Activity of Some Species of Genus Scutellaria Growing in Bulgaria

**DOI:** 10.3390/plants10010045

**Published:** 2020-12-28

**Authors:** Yoana Georgieva, Mariana Katsarova, Plamen Stoyanov, Rumen Mladenov, Petko Denev, Desislava Teneva, Evgeniy Plotnikov, Petko Bozov, Stela Dimitrova

**Affiliations:** 1Department of Pharmacognosy and Pharmaceutical Chemistry, Faculty of Pharmacy, Medical University of Plovdiv, 15A Vassil Aprilov, 4002 Plovdiv, Bulgaria; Yoana.Georgieva@mu-plovdiv.bg; 2Department of Bioorganic Chemistry, Faculty of Pharmacy, Medical University of Plovdiv, 15A Vassil Aprilov, 4002 Plovdiv, Bulgaria; Mariana.Katsarova@mu-plovdiv.bg (M.K.); plamen.stoyanov@mu-plovdiv.bg (P.S.); rummlad@uni-plovdiv.bg (R.M.); 3Laboratory of Biologically Active Substances, Institute of Organic Chemistry with Centre of Phytochemistry—BAS, 139 Ruski, 4000 Plovdiv, Bulgaria; petko.denev@orgchm.bas.bg (P.D.); dteneva@orgchm.bas.bg (D.T.); 4Research School of Chemistry and Applied Biomedical Sciences, Polytechnic University, 30 Lenin, 634050 Tomsk, Russia; plotnikov.e@mail.ru; 5Department of Biochemistry and Microbiology, Faculty of Biology, University of Plovdiv Paisii Hilendarski, 24 Tzar Asen, 4000 Plovdiv, Bulgaria; bozov@uni-plovdiv.bg; 6Research Institute, Medical University of Plovdiv, 15A Vassil Aprilov, 4002 Plovdiv, Bulgaria

**Keywords:** *Scutellaria*, antioxidant activity, baicalin, carbohidrates, organic acids, scutellarin, wogonoside

## Abstract

Until now, the interest to plants from genus *Scutellaria* in Bulgaria has been focused mainly on the terpenes in them. The purpose of this study is to enrich the information on the composition of the Bulgarian *Scutellaria* species in terms of both polyphenolic content as well as primary metabolites such as mono-, oligosaccharides and organic acids. An aerial part of three *Scutellaria* species growing in four low mountain regions of Southern Bulgaria was used. The flavonoids scutellarin, baicalin, baicalein, wogonin, wogonoside, luteolin, chrysin and a caffeoyl phenylethanoid glycoside-verbascoside have been identified via HPLC in different extracts from *Scutellaria altissima, Scutellaria albida* and *Scutellaria galericulata*. The antioxidant activity of the extracts has been evaluated. The *Scutellaria altissima* from Mezek and *Scutellaria galericulata* from Parvenets we studied, which are the richest in flavonoids (represented mainly by baicalin, scutellarin and wogonoside), show the highest Oxygen Radical Absorption Capacity. Hydroxyl Radical Averting Capacity of *Scutellaria albida* from Mezek and *Scutellaria altissima* from Bachkovo is the most pronounced, probably due to the content of scutellarin and luteolin and chrysin, respectively. Antioxidant activity of aqueous, methanolic and 70% and 96% ethanol extracts were also determined by the electrochemical method.

## 1. Introduction

The genus *Scutellaria* belongs to one of the most widely used in traditional medicine and is the subject of many phytochemical studies of the plant family Lamiaceae. About 350 species are known to be common in East Asia, Europe and North America [1]. *Scutellaria baicalensis* is one of the most well-known and best-studied species, the roots of which have been used in traditional Chinese medicine for millennia [2]. Other species of this genus have also been used by many cultures for treatment of hypertension, atherosclerosis, inflammatory diseases, and have shown sedative, antioxidant, antimicrobial, anxiolytic, insecticidal, and antiviral properties [1]. This activity has been proven to be mainly due to secondary metabolites, which are synthesized by plants for a protective purpose [2,3]. Such are flavonoids [4], phenolic acids [5], phenylethanoid glycosides [6] and terpenes (neo-clerrodanes [7], iridoids [8]) and act both individually and in combination. However, information on the composition of primary metabolites (sugars, organic acids, tocopherols [9] responsible for the development of the plants themselves is scarce. Organic acids are involved in several biochemical pathways, including energy production and formation of precursors for aminoacid biosynthesis [10]. Mono- and oligosaccharides, with low molecular weight, and their derivatives display a major role in the structure and function of the living cells [11]. They have been shown to affect primary and secondary metabolism, development and gene expression [12]. In their study, Park et al. demonstrated the influence of various carbohydrate sources on the type and amount of flavonoids accumulated in hairy root cultures of *Scutellaria baicalensis* [13]. The main group of biologically active compounds is undoubtedly the flavonoids and their glycosides [2]. They are natural antioxidants which act by neutralizing reactive oxygen species and chelating transition metal ions [14]. It is well documented that oxidative stress is involved in the etiology of a large number of human diseases, including diabetes, atherosclerosis, ischemia, neuropathological disorders such as Parkinson’s and Alzheimer’s disease, as well as the aging process [15] and natural antioxidants are being considered as prospective therapeutic agents.

In the flora of Bulgaria, genus *Scutellaria* is represented by 8 species: S. *albida*, S. *alpina*, S. *altissima*, S. *columnae*, S. *galericulata*, S. *hastifolia*, S. *orientalis* ssp. *pinnatifida*, S. *velenovskyi* [16]. Until now, the interest has been focused mainly on the terpenes in them. Neo-clerodane diterpenoids [17,18] were isolated and their antimicrobial [19] and antifeedant [20] activity demonstrated.

The purpose of this study is to enrich the information on the composition of the Bulgarian *Scutellaria* species in terms of both total polyphenolic and flavonoid content as well as primary metabolites such as mono- and oligosaccharides and organic acids. An aerial part of three *Scutellaria* species was used: *S. albida, S. altissima* and *S. galericulata*, growing in four low mountain regions of Southern Bulgaria. A comparative study of the composition of aqueous, methanol and ethanol extracts of the above *Scutellaria* species was made. The species-specific scutellarin, baicalin, baicalein, wogonin, wogonoside have also been identified and the antioxidant activity (AOA) of the extracts has been evaluated by three methods: Oxygen Radical Absorption Capacity (ORAC), Hydroxyl Radical Averting Capacity (HORAC) and electrochemical method.

## 2. Results

In the present study, a comparative research has been performed of the amounts of secondary (polyphenols and flavonoids) and primary (organic acids and carbohydrates) metabolites in the aerial part of three species of *Scutellaria* growing in four lowland regions of Southern Bulgaria as follows: *Scutellaria altissima* from the areas of Mezek(M) and Bachkovo(B), *Scutellaria albida* from Mezek(M) and Asenovgrad(A) and *Scutellaria galericulata* from Parvenets(P). It is known that the biological activity of plants of this genus is mainly due to the phenolic compounds synthesized by them [1,4]. Table 1 presents our results, which show that there is a correlation (*r* = 0.96) between the obtained amounts of total polyphenols and total flavonoids. The total polyphenolic content varies from 1353.1 ± 33.6 in S. *albida*(A) to 3498.5 ± 61.6 mg GAE/100 g dry *wt* in S. *altissima*(M), and the total flavonoid content from 162.0 ± 4.3 to 747.2 ± 5.8 mg QE/100 g dry *wt* in S. *albida*(A) and in S. *galericulata*(P), respectively. The amounts of individual flavonoids characteristic of the genus *Scutellaria* and extracted with different solvents are given in Figure 1.

Their extraction was performed with water, 70% ethanol, 96% ethanol and methanol. The most complete extraction of all flavonoids is achieved with 70% ethanol. Scutellarin is present in all samples with the highest amount in plant material of S. *albida*(M) (4350 µg/g) and S. *altissima*(M) (4219 µg/g) followed by S. *altissima*(B) (3263 μg/g) and S. *albida*(A) (3075 μg/g). Baicalin is contained in four of the five samples and its amount varies in a very wide range from 363 to 605 µg/g in S. *albida*(M) and S. *albida*(A) to 23,125–31,250 µg/g in S. *galericulata*(P) and S. *altissima*(M). Wogonoside is found in the plant material of all studied species of *Scutellaria* from 419 to 2827 µg/g, as well as wogonin, but in small quantities. However, wogonin is not found in S. *altissima*(B). Baicalein is found mainly in S. *galericulata*(P) (1047 µg/g) and in S. *altissima*(M) (288 µg/g). Verbascoside is present in the aerial part of S. *albida*(A), S. *albida*(M) and S. *galericulata*(P) and is best extracted with methanol.

The amounts of primary metabolites such as some mono-, disaccharides and organic acids have also been determined. Table 2 presents results for the content of fructose, glucose, galactose, rhamnose, xylose, fucose, sucrose, maltose and cellobiose, and Table 3 presents those of quinic, malic, ascorbic, citric, α-ketoglutaric, succinic, oxalic and tartaric acids. S. *albida*(M) shows the highest total carbohydrate content (8290.8 mg/100g dry *wt*), while S. *albida*(A) contains a significant amount of organic acids (9397.7 mg/100g dry *wt*).

Glucose is the main representative of carbohydrates. It is contained in the largest amount up to 2100 mg/100 g dry *wt* in plant material from S. *albida* from Mezek and Asenovgrad as well as S. *altissima* from Mezek and Bachkovo. Of the acids, this is succinic acid, which is found in the largest amount in S. *albida* (A), S. *albida* (M) and S. *altissima* (M) in the range from 4823.2 ± 124.8 to 1803.7 ± 114.1 mg/100 g dry *wt*.

Three methods were applied to determine the AOA of extracts from the studied *Scutellaria* species, namely ORAC, HORAC and electrochemical one. S. *altissima*(M) and S. *galericulata*(P) show the highest activity according to ORAC (1280.2 and 1155.2 µmol TE/g), S. *albida*(M) and S. *altissima*(B) (653.8 and 583.0 µmol GAE/g) according to HORAC (Table 4) and the methanol extract of S. *albida*(M) (16.1 AOA) according to the electrochemical (Table 5).

## 3. Discussion

The qualities of medicinal plants are determined by the combination of active substances that they synthesize. This in turn depends on both the species itself and environmental factors such as temperature, light condition, and water status [21]. Regarding the influence of these factors on the flavonoid composition of plants from the genus *Scutellaria*, the information in the available literature is mainly for species *baicalensis* [22,23,24]. *Scutellaria* species growing in different areas at the foot of the Rhodope Mountains have been selected for the present study. It is noteworthy that *S. altissima* from both regions Mezek and Bachkovo and *S. galericulata*(P) show a higher total amount of polyphenols and flavonoids compared to *S. albida* (Table 1). The amount of polyphenols in the order of about 3000 mg GAE/100 g dry plant material obtained by us is similar to the results of Karolak et al., who studied cultured *S. altissima* [6]. However, they detect twice as many flavonoids as us. Higher total polyphenolic and flavonoid content in the dry extract of *S. altissima* compared to *S. albida* was also found by Vaiday et al., who studied 16 *Scutellaria* species cultivated in Georgia [25]. Şenol et al. studied 33 *Scutellaria* species from Turkey and reported a similar relationship between the amount of these biologically active substances in dry extracts of *S. galericulata* and *S. albida* [26]. In support of this, our results for specific flavonoids (Figure 1) show that only *S. altissima*(M) and *S. galericulata*(P) contain baicalein and almost 100 times more baicalin than the other samples tested. Interestingly, we recorded 31,250 µg/g baicalin in the aerial part of *S. altissima*(M), while Karolak et al. found 22,570 µg/g in the roots and only 270 µg/g in the stems (shoots) from 2-year-old plants from *S. altissima* [6]. However, scutellarin is the one that is characteristic and present in all species studied by us, regardless of their habitat. In the plant material of *S. albida* from both regions, in addition to significant amounts of scutellarin, we also find the phenylethanoid glycoside-verbascoside, which was also identified by Matsa et al. [27] as well as luteoline—determined by Bardakci et al. [28] in *S. albida subsp velenovskyi* from the Karabük region of Turkey.

It is known that the AOA of plant extracts is due to phenolic compounds and especially to flavonoids in plant material. The *S. altissima*(M) and *S. galericulata*(P) we studied, which are the richest in polyphenols and flavonoids (represented mainly by baicalin, scutellarin and wogonoside), show the highest ORAC-assay AOA, respectively. This assay is based on the transfer of a hydrogen atom from the antioxidant to the peroxide radical. HORAC-assay takes into account the chelating ability of the antioxidant and according to it the AOA of *S. albida*(M) and *S. altissima*(B) is the most pronounced, probably due to the content of scutellarin and luteolin and chrysin, respectively. Baicalin, baicalein, and scutellarin are very strong antioxidants because the 6th position in nucleus A and 4’ in nucleus B, in their molecules, contain OH-groups that are thought to contribute most for radical scavenging ability. AOA of aqueous, methanolic and 70% and 96% ethanol extracts was determined by electrochemical method. The extracts obtained with 70% ethanol from all five samples contained the highest amount of biologically active substances and showed activity in the range from 7.9 to 10.1 AOA, again the highest being that of *S. altissima*(M). However, methanol extracts of *S. albida*(M), *S. galericulata*(P) and *S. albida*(A) have been shown to be more active. This is probably due to the presence of verbascoside, which is best extracted with methanol and has strong antioxidant properties [29]. Karolak et al. also identified verbascoside in methanol extracts from shoot culture, in vitro-regenerated and in vivo-derived plants of *S. altissima* and determined their AOA by ABTS, FRAP and LPO assays. The authors confirmed that the total polyphenolic content correlated with the AOA determined by the three methods [6]. Lohani et al. determine the antioxidant potential of aqueous and ethanolic extract of *S. lateriflora* in mouse brain tissue and demonstrate a significant effect [30].

It turns out that while there is plenty of information about the secondary metabolites in the plants of the genus *Scutellaria*, there is not much data about the primary metabolites in them. Olennikov et al. isolate and characterize polysaccharides from the aerial part of *S. baicalensis* Georgi, in which they prove galactose, arabinose and glucose [31]. In addition, they determine their antioxidant activity by two methods—β-carotene bleaching assay and the DPPH method. There are reports in the available scientific literature of free radical scavenging ability of polysaccharides from fungi, plants and algae [32,33,34]. Hernandes-Marin and Martinez confirmed this ability in their theoretical study on some mono-(d-glucose, d-fructose), di-(sucrose, maltose) and trisaccharides (1-kestose, 6-kestose, raffinose) [11]. Our study sheds light on the carbohydrate content of *Scutellaria* genus plants growing in Bulgaria. We find the monosaccharides glucose, fructose, galactose, rhamnose, xylose and the disaccharides sucrose, cellobiose in a very wide concentration range (Table 2). Fructose is present in the largest amount in *S. albida*(M), galactose—in *S. altissima*(B) and glucose—in *S. albida*(A). According to a study by Park et al., they are extremely important for the biosynthesis of flavonoids. They investigated the optimal carbohydrate source in hairy root cultures of *S. baicalensis* infected with *Agrobacterium rhizogenes* strain and proved that 150 mM sucrose in the culture medium increases the production of baicalein, fructose causes the greatest accumulation of baicalin and galactose—of wogonin [13].

It is important to know the composition of carbohydrates and organic acids, which are consumed directly, as infusions of plants can be easily prepared at home. As for the content of organic acids in the plants of the genus *Scutellaria*, again the data are only for *S. baicalensis*. Chirikova et al. monitor the accumulation of organic acids during the vegetation of the plant. They identified tartaric, citric, malic, malonic, succinic and fumaric acids in the aerial part of *S. baicalensis* Georgi and found that citric acid predominates and is detected in all phases of plant development [34]. The largest amount citric acid we find in *S. altissima*(M). In contrast, we also determine the amounts of quinic, oxalic, α-ketoglutaric and ascorbic acids. *S. albida*(A) contains the most succinic and quinic acids −4.8 and 2.9 %, respectively, and they have the largest contribution to the total amount of organic acids of 9.4% in it. It is highly likely that the ascorbic acid contained in *S. altissima*(M) and *S. altissima*(B) contributes to the pronounced AOA of these two species.

The climatic specifics of the area where the medicinal plants grow inevitably affect the synthesis of biologically active substances. It is important to note that the flavonoids scutellarin, baicalin, baicalein, wogonin, wogoniside and a caffeoyl phenylethanoid glycoside - verbascoside, responsible for the therapeutic action of *Scutellaria baicalensis* (Baikal skullcap) and *Scutellaria lateriflora* (American skullcap) are found for the first time in species of the genus *Scutellaria* growing in Bulgaria. A consequence of the high polyphenolic and flavonoid content is the pronounced AOA of the studied species. Based on the shown antioxidant activity, pharmacological effects of extracts of some of the studied species can be sought, related to the reduction of oxidative stress. In order to obtain a more complete phytochemical profile of the studied *Scutellaria* species, in addition to determining the amounts of secondary metabolites contained in them, a comparative study of some primary metabolites, such as organic acids and carbohydrates, was performed.

## 4. Materials and Methods 

### 4.1. Chemicals

Standards of flavonoids (scutellarin, baicalin, baicalein, wogonin, wogonoside, luteolin, chrysin), verbascoside, sugars (glucose, fructose, xylose, galactose, rhamnose, sucrose and cellobiose) and organic acids (quinic, malic, ascorbic, succinic, citric, α-keto-glutaric, oxalic and tartaric), methanol and acetonitrile (HPLC gradient grade) were purchased from Sigma-Aldrich (Darmstadt, Germany). Water was obtained from Milipore Milli-Q Gradient water purification system (Barnstead, US).

### 4.2. Plant Material

Aerial parts from three species of genus *Scutellaria* were collected during flowering in June 2017: *Scutellaria altissima* from the areas of Mezek and Bachkovo, *Scutellaria albida* from Mezek and Asenovgrad and *Scutellaria galericulata* from Parvenets. Collected raw materials were dried at 25 °C and powdered. Plant materials were authenticated by prof. Rumen Mladenov. Voucher specimens for *Scutellaria altissima* (n.062641), *Scutellaria albida* (n.062861) and *Scutellaria galericulata* (n.062642) were deposited at the Herbarium of the University of Agriculture, Plovdiv, Bulgaria.

### 4.3. Extraction of Polyphenols 

Approximately 0.5 g of the dried powders were weighted accurately, transferred to extraction tubes and mixed with 40 mL of the extragent (60% acetone solution in 0.5% formic acid) [35]. The extraction was conducted on an orbital shaker at room temperature for one h. Afterward, the samples were centrifuged (6000× *g*) and supernatants were further used for antioxidant activity determination and analysis of total polyphenols and flavonoids.

### 4.4. Determination of Total Phenolic/Flavonoid Contents

The total polyphenols were colorimetrically determined with the Folin–Ciocalteu reagent according to the method of Singleton et al. [36]. Gallic acid was employed as a calibration standard and the results were expressed as mg gallic acid equivalents (GAE) per 100 g dry weight.

The total flavonoid content was determined with AlCl_3_ reagent according to Chang et al. [37]. The calibration curve was constructed with quercetin dihydrate (10–200 mg/L). The results are expressed as mg quercetin equivalents (QE) per 100 g dry weight.

### 4.5. Extraction and HPLC Analysis of Individual Flavonoids

Two percent solutions of plant material in distilled water, 70% ethanol, 96% ethanol and methanol, respectively, were prepared. The extraction was performed via maceration at room temperature 25 °C for 24h. The obtained extracts were filtered through a microfilter (0.25 μm) and injected into the HPLC system. Verbascoside, scutellarin, baicalin, baicalein, wogonin, wogonoside, luteolin and chrysin were determined using the previously developed and validated HPLC method [38] on HPLC system (Varian, Australia) equipped with a ProStar230 solvent delivery module, photodiode array detector model 335 and HitachiC18 AQ (250 mm × 4.6 mm, 5 μm) column. The results are given as µg/g dry weight.

### 4.6. Extraction of Carbohydrates and Organic Acids

One gram of the powders was weighted accurately and subjected to extraction with 30 mL 3% meta-phosphoric acid in distilled water for 1 h at 30 °C and shaking on thermostatic water bath (NÜVE, Turkey). Afterward, the samples were centrifuged (6000× *g*) and the supernatants were used for HPLC analysis of sugars and organic acids. 

### 4.7. HPLC Analysis of Carbohydrates/Organic Acids

HPLC determination of glucose, fructose, xylose, galactose, rhamnose, sucrose and cellobiose was performed on Agilent 1220 HPLC system (Agilent Technology, USA), equipped with binary pump and Refractive Index Detector. Separation was performed using Aminex HPX-87H column (300 mm × 7.8 mm, BioRad), eluent 4 mM H_2_SO_4_, flow 0.5 mL/min, temperature 25 °C. Results are expressed as mg/100 g dry weight. 

HPLC determination of quinic, malic, ascorbic, succinic, citric, α-keto-glutaric, oxalic and tartaric acids was performed on Agilent 1220 HPLC system (Agilent Technology, USA), equipped with binary pump and UV-Vis detector. Wavelength of 210 nm was used. Organic acid separation was performed using Agilent TC-C18 column (250 mm × 4.6 mm, 5 μm) at 25 °C. The mobile phase was 25 mM phosphate (K_2_HPO_4_/H_3_PO_4_) buffer (pH 2.4), flowing at 0.8 mL/min. Results are expressed as mg/100 g dry weight.

### 4.8. Determination of Antioxidant Activity

#### 4.8.1. Oxygen Radical Absorbance Capacity (ORAC) Assay 

The method developed by Ou et al. was used with some modifications [39]. This method measures the ability of an antioxidant to neutralize peroxid radicals. It is based on the inhibition of the decline of fluorescence of fluorescein during its oxidation in the presence of an antioxidant. The thermal decomposition of 2,2’-azobis(2-amidinopropane) dihydrochloride (AAPH) is used as a peroxid radical generator. The results are expressed in μmol Trolox equivalents per gram of extract. Measurements are performed on FLUOstar OPTIMA fluorometer (BMG LABTECH, Offenburg, Germany). The excitation wavelength of 485 nm and emission wavelength of 520 nm were used. 

#### 4.8.2. Hydroxyl Radical Averting Capacity (HORAC) Assay 

The method was developed by Ou et al., and measures the ability of an antioxidant to form complexes in conditions of Fenton reaction, caused by the interaction between Co(II) and H_2_O_2_ [40]. The results are expressed in μmol gallic acid equivalents per gram of extract. Measurements are performed on FLUOstar OPTIMA fluorometer (BMG LABTECH, Offenburg, Germany). The excitation wavelength of 485 nm and emission wavelength of 520 nm were used. 

#### 4.8.3. Electrochemical Method for Determination of AOA

The electrochemical method was used to determine the AOA [41]. The experiment’s methodology consists in taking voltamperogram of cathodic electroreduction of oxygen using the “Analyst AOA” (RU.C.31.113.A N28715), connected to a PC. The AOA of the tested samples (extracts obtained with different solvents as described in 4.5.) was calculated according to kinetic criterion K (in micromoles per litre·minute) indicating the quantity of the reactive oxygen species in time, compared to the trolox kinetic criterion and expressed as: (1)$$AOA=\frac{K_{sample}}{K_{trolox}}$$

### 4.9. Statistics

The processing was repeated two times and the analysis performed at least in triplicate. Results were expressed as mean values ± standard deviations. When needed, statistical comparisons were made using Duncan’s multiple range test and Pearson correlation coefficient (*r*) was used to express correlations. A *p*-value of < 0.05 was taken to be significant. Statistical analysis was carried out using IBM SPSS 17.0.

## Figures and Tables

**Figure 1 plants-10-00045-f001:**
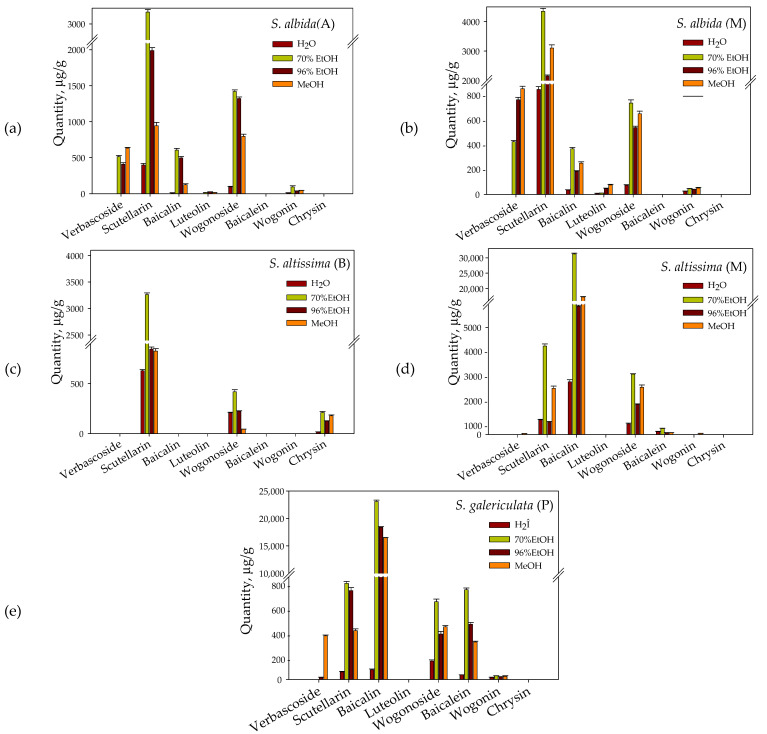
Content of Secondary Metabolites Determined by HPLC in Extracts from: (**a**) S. *albida* (A); (**b**) S. *albida* (M); (**c**) S. *altissima* (B); (**d**) S. *altissima* (M) and (**e**) S. *galericulata* (P) obtained with Different Solvents.

**Table 1 plants-10-00045-t001:** Content of Phenolic and Flavonoid Compounds in *Scutellaria* Species.

***Scutellaria* Species**	**Total Contents**	
	**Polyphenols**, mg GAE ^1^/100g dry *wt*	**Flavonoids**, mg QE ^2^/100g dry *wt*
*S. altissima* (M)	3498.5 ^e^ ± 61.6	610.7 ^c^ ± 32.4
*S. galericulata* (P)	3256.5 ^d^ ± 87.8	747.2 ^d^ ± 5.8
*S. altissima* (B)	2048.1 ^c^ ± 31.6	296.4 ^b^ ± 24.9
*S. albida* (M)	1863.4 ^b^ ± 38.7	255.6 ^b^ ± 10.9
*S. albida* (A)	1353.1 ^a^ ± 33.6	162.0 ^a^ ± 4.3

Results are presented as mean values ± standard deviations. There are no significant differences among values marked with the same superscript letters in individual columns; ^1^ GAE—gallic acid equivalent; ^2^ QE—quercetin equivalent.

**Table 2 plants-10-00045-t002:** Content of Carbohydrates in *Scutellaria* Species.

Carbohydrate,	*Scutellaria* Species				
mg/100g dry *wt*	*S. albida*(M)	*S. albida*(A)	*S. altissima*(B)	*S. altissima*(M)	*S. galericulata*(P)
Fructose	2160.5 ^d^ ± 173.1	1584.5 ^c^ ± 118.5	1095.1 ^b^ ± 89.5	1297.0 ^b^ ± 109.7	507.5 ^a^ ± 35.8
Glucose	2306.8 ^b^ ± 203.7	2319.7 ^b^ ± 213.7	2206.4 ^b^ ± 190.6	2148.7 ^b^ ± 190.8	901.4 ^a^ ± 65.1
Galactose	1137.2 ^b^ ± 83.7	1150.8 ^b^ ± 91.1	2238.0 ^c^ ± 183.8	1035.4 ^b^ ± 73.5	384.3 ^a^ ± 25.4
Rhamnose	203.2 ^c^ ± 12.3	152.5 ^b^ ± 9.3	254.9 ^d^ ± 15.5	127.7 ^a^ ± 6.7	157.9 ^b^ ± 8.5
Xylose	1264.7 ^d^ ± 106.5	1012.5 ^d^ ± 93.3	624.6 ^c^ ± 52.5	256.7 ^b^ ± 18.7	120.7 ^a^ ± 8.1
Sucrose	873.7 ^bc^ ± 77.4	1240.1 ^d^ ± 94.1	756.9 ^b^ ± 56.9	590.8 ^a^ ± 29.1	1018.8 ^cd^ ± 90.9
Cellobiose	344.6 ^a^ ± 24.5	337.3 ^a^ ± 13.7	546.9 ^b^ ± 34.7	806.5 ^c^ ± 50.6	458.1 ^b^ ± 27.8
^1^ Total	8290.8	7797.4	7722.8	6262.8	3548.7

Results are presented as mean values ± standard deviations. There are no significant differences among values marked with the same superscript letters in individual rows. ^1^ Total —sum of determined individual carbohydrates.

**Table 3 plants-10-00045-t003:** Content of Organic Acids in *Scutellaria* Species.

Acid,	*Scutellaria* Species				
mg/100 g dry *wt*	*S. albida*(A)	*S. altissima* (M)	*S. altissima*(B)	*S. albida* (M)	*S. galericulata*(P)
Quinic	2918.9 ^d^ ± 34.1	222.4 ^a^ ± 21.7	1724.4 ^c^ ± 120.3	413.4 ^b^ ± 28.6	203.1 ^a^ ± 20.6
Malic	538.9 ^b^ ± 26.6	120.7 ^a^ ± 10.5	113.8 ^a^ ± 12.0	112.4 ^a^ ± 17.0	115.3 ^a^ ± 11.8
Ascorbic	93.7 ^b^ ± 5.3	93.7 ^b^ ± 5.3	148.9 ^c^ ± 10.9	60.7 ^a^ ± 3.3	91.1 ^b^ ± 6.4
Citric	267.3 ^a^ ± 25.5	1029.1 ^c^ ± 57.1	615.9 ^b^ ± 28.6	621.3 ^b^ ± 46.1	608.5 ^b^ ± 53.7
α-Ketoglutaric	259.8 ^c^ ± 18.3	86.4 ^a^ ± 8.7	152.2 ^b^ ± 13.9	100.1 ^a^ ± 9.9	132.1 ^b^ ± 8.9
Succinic	4823.2 ^e^ ± 124.8	2538.5 ^d^ ± 152.8	929.3 ^b^ ± 35.9	1803.7 ^c^ ± 114.1	294.5 ^a^ ± 27.6
Oxalic	495.9 ^e^ ± 10.4	19.5 ^b^ ± 1.6	332.1 ^d^ ± 8.2	67.9 ± 3.7	8.9 ^a^ ± 0.2
Tartaric	n.d.	25.4 ^b^ ± 2.7	113.2 ^c^ ± 6.0	22.6 ^b^ ± 1.7	9.8 ^a^ ± 0.7
^1^ Total	9397.7	4148.4	4129.8	3220.1	1463.3

Results are presented as mean values ± standard deviations. There are no significant differences among values marked with the same superscript letters in individual rows; ^1^ Total —sum of determined individual organic acids; n.d.—not detected.

**Table 4 plants-10-00045-t004:** Oxygen Radical Absorption Capacity (ORAC) and Hydroxyl Radical Averting Capacity (HORAC) Antioxidant Activity of Extracts from *Scutellaria* Species.

***Scutellaria* Species**	**Antioxidant Activity**	
	ORAC, µmol TE ^1^/g	HORAC, µmol GAE ^2^/g
*S. altissima* (M)	1280.2 ^c^ ± 79.7	387.5 ^c^ ± 29.4
*S. galericulata* (P)	1155.2 ^b,c^ ± 119.2	302.4 ^b^ ± 27.2
*S. albida* (M)	1005.6 ^b^ ± 68.8	653.8 ^d^ ± 22.2
*S. altissima* (B)	926.9 ^b^ ± 66.2	583.0 ^d^ ± 32.1
*S. albida* (A)	652.3 ^a^ ± 51.9	201.3 ^a^ ± 12.4

Results are presented as mean values ± standard deviations. There are no significant differences among values marked with the same superscript letters in individual columns; ^1^ TE—trolox equivalent; ^2^ GAE—gallic acid equivalent.

**Table 5 plants-10-00045-t005:** Antioxidant Activity (AOA) of Extracts from *Scutellaria* Species, as Measured by Electrochemical Method.

*Scutellaria* Species	Extract	K, μmol/L per min ± SD	AOA
*S. altissima* (M)	H_2_O	17.000 ^c^ ± 1.009	5.8
	70% EtOH	29.392 ^b^ ± 2.043	10.1
	96% EtOH	22.824 ^b^ ± 2.134	7.8
	MeOH	31.481 ^ab^ ± 3.003	10.8
*S. albida* (A)	H_2_O	8.368 ^a^ ± 0.148	2.9
	70% EtOH	24.741 ^a^ ± 1.242	8.5
	96% EtOH	20.991 ^b^ ± 2.213	7.2
	MeOH	34.811 ^b^ ± 1.983	11.9
*S. albida* (M)	H_2_O	10.055 ^b^ ± 0.654	3.5
	70% EtOH	23.192 ^a^ ± 1.842	7.9
	96% EtOH	20.263 ^b^ ± 1.325	6.9
	MeOH	46.815 ^c^ ± 2.301	16.1
*S. galericulata* (P)	H_2_O	9.924 ^b^ ± 0.492	3.4
	70% EtOH	26.157 ^a^ ± 1.992	8.9
	96% EtOH	24.743 ^b^ ± 2.891	8.5
	MeOH	43.784 ^c^ ± 3.231	15.0
*S. altissima* (B)	H_2_O	7.163 ^a^ ± 1.016	2.5
	70% EtOH	22.981 ^a^ ± 1.194	7.9
	96% EtOH	22.706 ^b^ ± 1.891	7.8
	MeOH	29.450 ^a^ ± 0.990	10.1
Trolox	96% EtOH	2.911 ^a^ ± 0.010	1.0

Results are presented as mean values ± standard deviations. There are no significant differences among values marked with the same superscript letters in individual rows.

## Data Availability

The data presented in this study are available on request from thecorresponding author.

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
