# Peer review of "Metabolite Profile and Antioxidant Activity of Some Species of Genus Scutellaria Growing in Bulgaria"

_plants, 2020, doi:10.3390/plants10010045_

Round 1

Reviewer 1 Report

The manuscript entitled "Metabolite profile and antioxidant activity of some species of genus Scutellaria growing in Bulgaria" describes the content of some secondary and primary metabolites of three species of Scutellaria collected in different regions. In addition, there is an evaluation of the antioxidant activity due to the high content of phenolic compounds in the extracts. The study was properly carried out and shows the phytochemical and antioxidant profile of these plant species. The manuscript should be revised, according to suggestions. After these changes, the study can be published in the journal Plants.

Suggestions:

In line 59, correct the term "Altzheimer’s disease" for "Alzheimer’s disease";

In lines 61-62 the terms of gender and species must be in italics; Even in the other parts of the manuscript. Please review it completely!

In line 72, correct the word "Hyrdoxyl" for "Hydroxyl";

In line 73, add a noun to the word “electrochemical”;

Figure 1 b is partially cut. Please correct it.

Figures 1a-d should contain the names of the metabolites, similar to the presentation in Figure 1e.

The H2O writing should be corrected in Figure 1e;

Line 117-119 has different sizes than the letters. Please standardize according to the journal's recommendations.

Line 131: replace "light condition, water status" with "light condition, and water status";

Line 151: please confirm the term "ssp." in S. albida ssp velenovskyi. You wanted to write the subspecies term, "subsp."

Author Response

Thank you for the recommendations. All grammatical errors have been corrected. The additions made to the text are marked in red. All the terms of gender and species are in Italic, but probably after the transformation into PDF some of them change. Figure 1 has been adjusted according to your requirements.

Reviewer 2 Report

The details of my revision has been inserted directly to the attached pdf file.

Author Response

Thank you for the recommendations. The sentence from lines 43 to 46 is divided. Significant differences between results were included in all tables.

Reviewer 3 Report

       Manuscript entitled „Metabolite profile and antioxidant activity of some species of genus Scutellaria growing in Bulgaria” which was submitted to revision in Plants provide valuable and lacking information concerning primary and secondary metabolites in three Scutellaria species grown in chosen Bulgaria regions. The project was planned properly, the results are rather presented clearly. Some inaccuracies in the manuscript or suggestions are placed below:

  • Explain why only three species from 8 grown in Bulgaria was chosen by authors? Why galericulata is only from one region, since the others are from two regions?
  • Wherever in the text authors write about correlations, it should be calculated (Pearson`s correlation the authors mean?)
  • Explain the abbreviation M, P, B, A under all the tables.
  • Add the letters in all results (tables) expressed statistically significant differences between the samples.
  • 1 should be present on one page (e.g. two pictures side by side, not one below one). It enable clear presentation. Fig.1. b should be improved (not visible MeOH in the legend). Add to the Fig.1. title information that the secondary metabolites were determined by HPLC.
  • Line 92-105. The authors gave the results of polyphenolic content in the range form… to… But there is no information in which solvent used for extraction.
  • Line 102. Wogonoside or wogonin is not found in S. altissima? All of the species should be written by italics.
  • Tables 2 and 3. In the brackets of the total of carbohydrate and organic acids should be add “sum of determined individual …”
  • In all the tables and figure there is no chronology in the analyzed species (e.g. Table 1. M,P,B,M,A, at the fig.1. A,M,B,M,P, etc.)
  • Line 137, correlates (as mentioned above).
  • Lines 158-160. The Pearson`s correlation perhaps confirm this statement.
  • In the discussion should be added information why some of the polyphenolic compounds are better or worse extracted with different solvents used, and why the same species (e.g. S. altissima) gave completely different results.
  • Line 226. polyphenols are a wide range of organic compounds and indeed include flavonoids.
  • Lines 227-231. Why this extragent was used for polyphenols extraction? The extraction process was carried out in only one repetition?
  • Lines 240-246. Give more details about the condition of extraction. It was shaken? Add the information how the results were expressed.
  • Lines 280-284. Add the information that the AOA was determined in four different extracts (four extragent).
  • Add the correlation analysis.

Author Response

 Manuscript entitled „Metabolite profile and antioxidant activity of some species of genus Scutellaria growing in Bulgaria” which was submitted to revision in Plants provide valuable and lacking information concerning primary and secondary metabolites in three Scutellaria species grown in chosen Bulgaria regions. The project was planned properly, the results are rather presented clearly. Some inaccuracies in the manuscript or suggestions are placed below:

  • Explain why only three species from 8 grown in Bulgaria was chosen by authors? Why galericulatais only from one region, since the others are from two regions?

We have chosen these three types of Scutellaria because they are the most spread. We have conducted additional research based on which we have an idea to develop a phytopreparation and it is important that the plant material is available.

  • Wherever in the text authors write about correlations, it should be calculated (Pearson`s correlation the authors mean?)

Indeed, Pearson correlation coefficient (r) was used to express correlations and it was included in the Statistics description. Corresponding correlation coefficients (r) were given in the text.

  • Explain the abbreviation M, P, B, A under all the tables.

The abbreviations M, P, B, A are entered at the beginning of the results section. We believe that if their explanation is repeated under each table, it would make it unnecessarily difficult.

  • Add the letters in all results (tables) expressed statistically significant differences between the samples.

It was done.for all tables.

  • 1 should be present on one page (e.g. two pictures side by side, not one below one). It enable clear presentation. Fig.1. b should be improved (not visible MeOH in the legend). Add to the Fig.1. title information that the secondary metabolites were determined by HPLC.

Figure 1 has been adjusted according to your requirements.

  • Line 92-105. The authors gave the results of polyphenolic content in the range form… to… But there is no information in which solvent used for extraction.

Extraction was performed with acetone as described in section 4.3. This solvent was used according to our previous research. A reference has also been added.

  • Line 102. Wogonoside or wogonin is not found in S. altissima? All of the species should be written by italics.

The sentence has been corrected. "It" has been replaced by "wogonin" to be clear.

All the terms of gender and species are in Italic, but probably after the transformation into PDF some of them change.

  • Tables 2 and 3. In the brackets of the total of carbohydrate and organic acids should be add “sum of determined individual …”

It was done.

  • In all the tables and figure there is no chronology in the analyzed species (e.g. Table 1. M,P,B,M,A, at the fig.1. A,M,B,M,P, etc.)

We have placed the results in descending order in all tables and figure. In our opinion, this is an appropriate way to focus on the highest concentration of a particular substance and the corresponding species of Scutellaria that synthesizes it.

  • Line 137, correlates (as mentioned above).

Here we show that our results for the total amount of polyphenols are similar to those obtained by other authors. We replace “correlates with” with “similar to”.

  • Lines 158-160. The Pearson`s correlation perhaps confirm this statement.

We change the sentence:

„Baicalin, baicalein, and scutellarin are responsible for the high AOA determined by these methods, as in their molecules the 6th place in nucleus A and 4' in nucleus B contain OH-groups that are thought to contribute most for radical scavenging ability“

 like this:

 “Baicalin, baicalein, and scutellarin are very strong antioxidants because the 6th position in nucleus A and 4' in nucleus B, in their molecules, contain OH-groups that are thought to contribute most for radical scavenging ability.”

  • In the discussion should be added information why some of the polyphenolic compounds are better or worse extracted with different solvents used, and why the same species (e.g. S.altissima) gave completely different results.

In addition to the species itself, environmental factors are also crucial for the biosynthesis of active substances [21]. Both species of Scutellaria inhabit lowland areas of southern Bulgaria, but while Scutellaria altissima from the region of Mezek grows in a sunny and stony place, that from Bachkovo in a shady and humid place. That was interesting for us too.

  • Line 226. polyphenols are a wide range of organic compounds and indeed include flavonoids.

The word flavonoids has been deleted.

  • Lines 227-231. Why this extragent was used for polyphenols extraction? The extraction process was carried out in only one repetition?

This solvent was used according to our previous research. A reference has also been added.

  • Lines 240-246. Give more details about the condition of extraction. It was shaken? Add the information how the results were expressed.

The extraction was performed by maceration. This is added in section 4.5. as well “The results are given as µg/g dry weight”.

  • Lines 280-284. Add the information that the AOA was determined in four different extracts (four extragent).

The same extracts in which the individual flavonoids were found were used to determine the AOA. This information is added in section 4.8.3.

  • Add the correlation analysis.

Pearson correlation coefficient (r) was used to express correlations and it was included in the Statistics description.

 The additions and corrections made to the text are marked in red.

Round 2

Reviewer 3 Report

All of the suggestions have been considered and are included.